# Sentiment of Chinese Tourists towards Malaysia Cultural Heritage Based on Online Travel Reviews

Zheng Cao [1,2], Heng Xu [1] and Brian Sheng-Xian Teo [2,*]

1   School of Management, Henan University of Technology, Zhengzhou 450001, China
2   Graduate School of Management, Management & Science University,
    Shah Alam 40100, Selangor Darul Ehsan, Malaysia
*   Correspondence: brian_teo@msu.edu.my

**Abstract:** Analyzing the perception differences and influencing factors of cross-cultural groups in heritage tourism can help heritage sites to formulate differentiated service and improve tourist satisfaction. This research adopted the BERT model to undertake sentiment analysis of 17,555 Chinese online reviews for nine scenic spots in Melaka. Using vocabulary filtering, co-occurrence analysis, and semantic clustering technology, the emotional characteristics of Chinese outbound tourists when they visited heritage sites in Melaka were analyzed, which revealed the factors influencing their positive and negative emotions. Results showed that: 1. The BERT-based deep learning approach can obtain improved sentiment predictive performance. 2. Chinese tourists' general emotional perceptions of Melaka were positive and they were very interested in heritage sites. 3. The most important reason for the negative emotions of Chinese tourists was a lack of cultural experience in Melaka. This research expands the application of deep learning in the field of tourism, and it helps heritage tourism destinations to improve their marketing plans for Chinese tourists and achieve long-term sustainable development of the destination.

**Keywords:** heritage tourism; sentiment analysis; BERT model; Chinese outbound tourists

## 1. Introduction

Heritage tourism is a rapidly growing and specialized tourism genre. The United Nations World Tourism Organization (UNWTO) (2015) considers heritage tourism "a key element of international tourism consumption" [1]. An increasing number of tourists visit places to enjoy a unique or distinctive heritage. Timothy (2011) even mentioned that approximately 85% of the general population is considered to be current or potential heritage tourists [2]. This significant tourist market attracts interest from both the academic and commercial sectors because of its multifaceted potential to generate revenue and sustain cities and ecologies [3]. The growing competition among tourism destinations and changes in tourists' expectations and habits are forcing destinations to find new ways of attracting tourists in order to stay competitive [4]. Image is the vitality of a tourist destination, which directly affects the decision-making behavior and satisfaction of tourists. Therefore, destination image and especially the factors influencing it are becoming extremely important for good positioning of destinations in the international tourism market, as well as for increasing their competitive advantages [4]. Analyzing tourists' perceptions of heritage sites will help heritage sites focus on tourists' experience feedback, develop new kinds of tourism experience activities, and improve service facilities and equipment, so as to meet the needs of tourists for a higher level of heritage sites, and continuously improve tourists' satisfaction [1]. Meanwhile, the analysis of heritage image can enrich the study of group differences in tourism destination image perception, to provide a theoretical basis for the precise marketing of heritage tourism, which is of great significance for the recovery and development of heritage tourism after the epidemic [5].

Therefore, we must gain a better understanding of tourists' experiences and behavioral intentions at heritage sites [6].

China was the world's largest outbound tourism market in 2019. Three years of epidemic stagnation was a huge loss to the global tourism market. The China National Health Commission announced that after 8 January 2023, "Class B and Class B management" will be implemented for COVID-19 infection, and China's epidemic prevention and control has now entered a new stage [7]. Many countries and regions are now focusing on the recovery and development of China's outbound tourism market. Chinese outbound tourism can bring huge spending power to the world tourism economy and will greatly contribute to the recovery process of global tourism [7]. Therefore, this study focuses on the perceptions of Chinese tourists to assist heritage tourism destinations in improving their marketing plans for Chinese tourists, and better enhance the competitiveness of the destinations.

With the emergence of social networks and online platforms, an increasing number of Chinese tourists choose to use the Internet to share their tourism experiences and spread tourism information [8]. This provides a vast new data source for perception analysis of tourist destinations. Mining online text information, and using sentiment analysis tools to identify positive or negative emotions expressed in the text, captures the feelings and feedback of individual tourists, helps provide more information about the destination image as tourists perceive it, and helps to improve destination quality, ensuring a balance among economic, social, and environmental issues and achievement of sustainable development [9]. Most existing sentiment analysis methods use machine learning algorithms, such as the maximum entropy classifier (Max. Ent.), naïve Bayes, and support vector machines (SVM) [10]. In recent years, deep learning techniques have gained popularity because they can improve the classification accuracy of data, particularly when the number of labeled data is thousands of examples.

Therefore, the purpose of this study was to analyze online travel reviews using a deep learning approach to investigate Chinese tourists' perceptions of the heritage site in Melaka, to reveal the emotional characteristics of Chinese outbound tourists and the factors influencing positive and negative emotions. This paper is organized as follows. Section 2 reviews related research on heritage tourism and sentiment analysis. Section 3 proposes a deep learning method for sentiment analysis based on the BERT model. Section 4 applies our methodology to sentiment analysis for Melaka. Section 5 discusses the results. Finally, conclusions and implications are presented.

## 2. Literature Review

### 2.1. Tourist Perception in Heritage Tourism

Perception theory was first used in the field of psychology. "Perception" is the direct reflection of objectivity by the sensory organs of the human brain, including sensation and perception. Tourist perception is an extension of perception theory in the field of tourism. It is the process of tourists' psychological cognition and transformation of tourism objects, tourism environment, and other information through their senses. The main content of perception includes the resources, environment, management, and services of the destination [11]. The interaction between tourism and heritage resources constitutes the heritage tourism experience [12]. As the core and main body of tourism activities, tourists and their psychological needs, behaviors, and experiences have important research significance for cultural heritage tourism. According to the existing literature, research on tourists' perception of heritage sites mainly focuses on the perceptual dimension, perceptual content, perceived value, perception-influencing factors, perceptual evaluation methods, perceived differences in different types of destinations, and perception differences between different groups. For example, Medina (2019) conducted an empirical study on the relationship between tourism motivation, tourists' perceptions, and tourists' satisfaction by considering tourists in Uveda and Baessa, world heritage sites in Spain, and found that tourists' cultural motivation was significantly and positively correlated with satisfaction [13]. Vong (2013) studied how tourists' views on Macao heritage management affected their perceptions

of and satisfaction with visiting heritage sites through a questionnaire survey [14]. Tian et al. (2021) studied the relationship between perceived value, tourist satisfaction, and the loyalty of tourists at the Summer Palace in Beijing using survey questions [15]. Pavli (2020) examined how travelers' perceptions of World Heritage Sites (WHS) differed by type of traveler using a questionnaire survey [4].

Most of this literature relies on traditional research methods, such as focus group interviews and questionnaires, which may have drawbacks such as sampling problems and self-reporting bias [16]. With the recent development of the Internet, most scholars have studied tourists' perceptions and experiences of heritage tourism using online text content. Kim et al. (2016), using the heritage site "Jeju island" as the keyword in a travel platform, collected 332 blog articles from different countries in the East and West, comparing Eastern and Western tourists with distinct personality differences [17]. Liu et al. (2017) used online comments as the data source with a content analysis method to study tourists' value perception of heritage tourism, and divided heritage tourists into various types according to differences in value perception [18]. Bai et. al. (2016) used grounded theory to extract five main categories and 32 subcategories for perceptions of the Terracotta Warriors of Qin Shihuang, based on tourist comments. They constructed a model for tourist perception evaluation based on three aspects: spatial structure, perception process, and perception content [19]. Wang (2019) summarized the tourist perception image into four dimensions: the spatial environment image of the scenic spot, tourism landscape image, tourism service image, and tourism experience cognitive image through the analysis of network text [20]. Although previous studies have discussed tourists' perceptions of heritage sites from multiple perspectives, they have not deeply analyzed tourists' emotional characteristics and have ignored the dynamic perceptions of tourists, which can be solved by sentiment analysis [21].

*2.2. Research on Tourist Sentiment Analysis*

Emotions are intuitive feelings triggered by situational cues, and they are highly socialized and developed from thought rather than instinctive emotions [22]. Using sentiment and emotional expressions to evaluate customer satisfaction and word-of-mouth has become a valuable alternative to traditional satisfaction research in marketing and consumer behavior disciplines [23]. To study tourism emotion factors and their possible meanings, researchers divide tourists' sentiments into positive and negative emotions as well as several basic types of emotions, such as love, happiness, regret, anger, sadness, and fear [24]. Hudson (2015) used structural equation modeling to examine the impact of social media interactions with travel brands on consumers' affective attachments [25]. Hosany (2011) determined that tourists' emotions, such as happiness, love, surprise, and fear, are triggered by evaluative factors such as goal congruence, internal commonality, interest goals, and personal imagination [12]. Prayag (2013) further identified different patterns of emotional responses by tourists, including happiness, non-emotion, passive, mixed, and passionate [26]. These studies further analyzed the emotions and sentiments of tourists during their travel, but they still relied on traditional methods, such as questionnaires and interviews. These methods are more appropriate for collecting structured and consistent types of information, but not for fragmented and unstructured UGC data [22].

With the development of big data analytics, tourism scholars are becoming increasingly interested in applying automated sentiment analysis to tourism data. In this study, sentiment analysis refers to using an automatic sentiment analysis model to automatically identify a large amount of text data containing tourism evaluations, classify them into positive and negative sentiments, and extract the sentiment characteristics of tourism [26]. Sentiment analysis can help researchers discover the dynamics of tourism perception based on large, interconnected datasets [27]. Recent travel research data sources using social media for sentiment analysis have focused on social networks such as Facebook, travel advisors, and Twitter. Related research has focused on the following aspects: sentiment analysis, travel preferences obtained from social media, social media communication strategies, identification of tourist attractions based on digital impressions using social media,

travel recommendation systems, cultural exposure to foreign cities through media, the definition of smart tourism, and current trends.

For example, Windasari et al. (2017) used emotion analysis and topic extraction techniques in their research to analyze the comments of Indonesian tourists on TripAdvisor [28]. Kim et al. (2017) used sentiment analysis techniques to analyze 19,835 comments from visitors to a website and inferred that traffic was the issue with the most critical negative comments for Paris [29]. Deraman (2021) obtained 11,357 pieces of data from Twitter and used emotional analysis to study tourists' views on tourism in Malaysia in the post-epidemic era [30]. Garner (2022) demonstrated that machine learning can help us better understand the concept of consumer well-being [31]. Correia (2020) used text mining, sentiment analysis, and market basket models with 8638 online travel reviews (2017–2018) to analyze the attraction of heritage sites for visitors [32]. Liu et al. (2020) analyzed 51,191 tourism reviews on websites such as Ctrip and TripAdvisor. It was found that domestic and foreign tourist perceptions of Macau's image as a destination differed. Zen (2021) used sentiment analysis to analyze 4183 TripAdvisor reviews by foreign tourists for the World Heritage Goreme National Park and Cappadocia Rocks to obtain results on tourism perceptions of cultural heritage [33].

### 2.3. Research on the Advances in Emotion Analysis Techniques

In the above literature, the sentiment analysis methods used are primarily dictionary-based and machine learning methods. The dictionary-based method makes category statistics and emotional value calculations based on emotional words in the text, which has the advantages of being easy to use and expand. However, sentiment classification methods based on sentiment dictionaries mainly rely on their construction of sentiment dictionaries. In this era of rapid Internet development, the speed of information updates has accelerated. Hence, many new words appear on the Internet regularly, so sentiment dictionaries are not very efficient in recognizing many new words, such as proverbs, idioms, or special network words. Sentiment dictionaries need to be expanded continuously to meet their needs, and the same emotion word may have different meanings when expressed in different times, languages, or domains; therefore, the method based on sentiment dictionaries is not ideal for cross-domain and cross-language analysis. Machine learning requires many previously labeled data as training samples to build classification models, and the standard models include naive Bayes (NB) and support vector machines (SVM). The limitation of the machine learning method is that when the training corpus is unclear, it will affect the results of the target text and training sample and cause the two deviations to be significant for achieving the ideal effect [34].

Deep learning-based sentiment analysis methods typically use neural networks such as convolutional neural networks (CNN), recurrent neural networks (RNN), and long short-term memory (LSTM) networks [35]. Deep learning methods have significant advantages in sentiment analysis compared to methods based on sentiment dictionaries and traditional machine learning. They can actively learn text features and retain the information of words in the text to better extract the semantic information of corresponding words and expressions and effectively categorize emotions based on their contents [36,37]. Pre-training language models make full use of a large-scale monolingual corpus and can model multiple meanings of a word. These models do not rely on manually annotated corpora, and they can automatically learn appropriate and practical features from words, grammatical information, and semantic similarities in the dataset while training the network, which makes deep learning models well suited for tasks such as sentiment analysis [38].

Recently, deep learning methods have been used in tourism management for related research and analysis. Li and Cao (2018) demonstrated that LSTM neural networks perform better than traditional methods at predicting tourism flows [39]. Li et al. (2019) used a deep learning approach to predict monthly visitor arrivals in Macau. They demonstrated that the deep learning approach significantly outperformed artificial neural network models such as support vector machines [40]. Zhang and Chen (2019) analyzed the visual content

of tourists' photos through an in-depth learning model and found tourists' behavior and perceptions of tourist destinations [41]. These studies demonstrated the superiority of deep learning. However, few studies have been conducted to analyze visitors' emotions using deep learning methods and to overcome the limitations of current sentiment analysis methods. Therefore, in this study, Chinese comments on a tourism platform were taken as the research data with which to investigate Chinese tourists' perceptions of Melaka through an emotion analysis method based on deep learning.

## 3. Methodology

Tourism websites are a popular form of social media, and an increasing number of Chinese tourists have expressed their views and feelings about their destinations and activities through travel platforms [42]. Considering the large number of variants in the Internet vocabulary, it is challenging to build a domain dictionary and select appropriate features. Therefore, this paper adopted a text sentiment classification model based on the bidirectional encoder representations from transformers (BERT) model, so as to better classify the online reviews emotionally.

BERT is a pre-training model for natural language processing (NLP) domain proposed by Devlin J et al. of Google in 2018 [43], which is based on an improvement of the Transformer model. The Encoder module in the bi-directional Transformer is used to build the model, and the input text information is extracted by this layer, which discards the circular structure of traditional NLP processing methods, such as RNN, LSTM, and other such models, and effectively solves the problem that the model cannot be processed in parallel and the long-term dependence of the text.

The core of the BERT model is the same as the Transformer model [43], whereby combining the connection between each word and other words in the text, removing the limitation of distance, and representing the dependency between the current word and the rest of the words in the sentence explicitly, the contextual information of the sentence is fully combined to better identify the semantic information of the sentence. Compared with word2vec, word meanings can be obtained based on sentence context, thus avoiding ambiguities. A deep learning method based on the BERT model for sentiment analysis is presented in Figure 1.

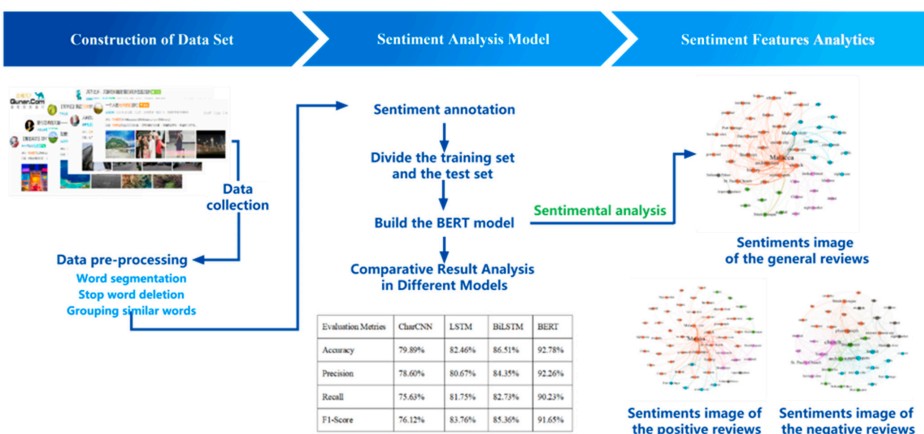

**Figure 1.** The BERT-based deep learning approach to tourism sentiment analytics.

The analytical process is divided into five steps: (1) data collection, (2) text preprocessing, (3) building the training set, (4) sentiment analysis based on the BERT model, and (5) precision inspection and analysis. The sample for this study consisted of review texts from Chinese tourism websites collected from Chinese tourists' reviews of the nine most popular attractions in Melaka. In the sentiment analysis phase, we used the BERT model for emotion classification and focused on vocabulary filtering, co-occurrence analysis, and semantic clustering technology to analyze online reviews written by Chinese tourists on a

domestic travel platform [44]. A complete explanation of the proposed model is provided in the following sections.

### 3.1. Study Area

Tourism has become the second-largest contributor to Malaysia's economic growth. China is Malaysia's third largest tourist source market. Over the past decade, Chinese tourists to Malaysia have increased by 200% from 0.94 million in 2008 to 2.94 million in 2018 [42]. In 2019, USD 2.84 billion was generated by Chinese tourists in Malaysia, accounting for 14% of the country's total tourism revenues [45]. Melaka is the second most popular city for Chinese travelling to Malaysia. Melaka is famous for its preserved heritage and culture. In July 2008, the United Nations Educational, Scientific, and Cultural Organization (UNESCO) listed Melaka as a World Heritage City. Therefore, it would be beneficial to select Melaka as the study area. We selected nine tourist attractions in Melaka: Stadthuys, the Melaka River, St. Paul's Church, Port Santiago, Jonker Street, Sultanate Palace, Maritime Museum, Dutch Square, and the Strait Mosque (Figure 2).

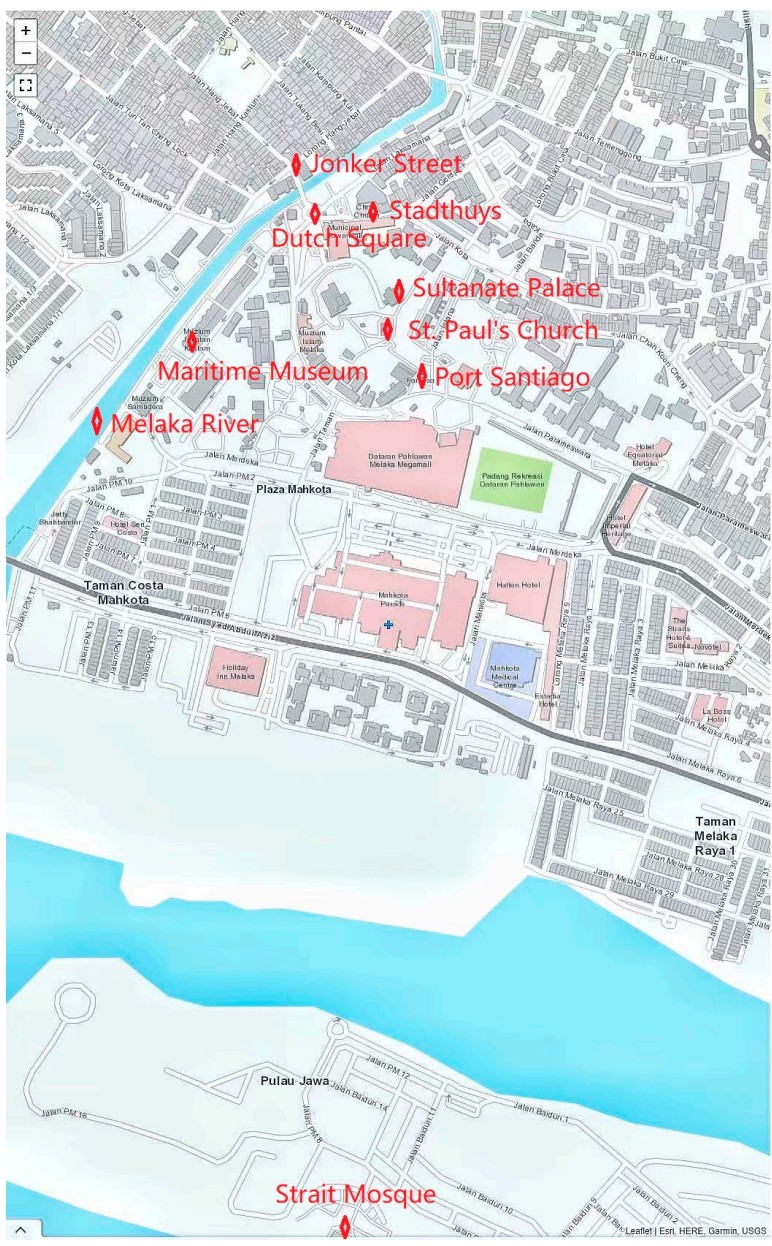

**Figure 2.** Study area. Note: Map from 23 January 2023, Earthol.com (https://www.earthol.com/dx/).

### 3.2. Data Collection

This study used the Chinese website Qunar (http://www.qunar.com, accessed on 1 December 2022). as the data source. Qunar.com is a leading online travel-information service platform in China. It includes travel notes, reviews, and other information on more than 60,000 tourism destinations worldwide. This website allows users to share their travel experiences, and the relevant information is updated regularly [46]. After comparing it with other well-known travel websites in China, we found that Melaka had the largest number of comments on Qunar's website, so we finally chose it for data mining. This study used Python's web crawler technology and collected 17,555 reviews voluntarily shared by Chinese tourists for the nine most popular scenic spots in Melaka as the initial data from 2015 to 2019 (Table 1). All countries restricted inbound and outbound travel in early 2020 in response to the epidemic [47]. Consequently, no new data were generated on visiting Malaysia in January 2020.

**Table 1.** Results of comments obtained for Melaka.

| Tourist Attraction | Frequency | Total % |
|---|---|---|
| Stadthuys | 4531 | 25.81% |
| Melaka River | 2707 | 15.42% |
| Jonker Street | 2457 | 14.0% |
| St. Paul's Church | 1803 | 10.27% |
| Port Santiago | 1615 | 9.19% |
| Dutch Square | 1369 | 7.80% |
| Strait Mosque | 1069 | 6.08% |
| Maritime Museum | 1026 | 5.85% |
| Sultanate Palace | 978 | 5.58% |
| Total | 17,555 | 100% |

### 3.3. Data Preprocessing

For preprocessing, first, the irrelevant and ambiguous data were removed. Second, the Python package "jieba" was used for Chinese word separation considering the accuracy and convenience of word separation. Third, Python software was used to remove invalid characters, thereby rendering the dataset clean and tidy.

### 3.4. Sentiment Analysis Model

In this task, first, all the sentences in the dataset were tagged using the BERT tokenizer to create a vocabulary table. Subsequently, we passed all tokens to the pre-trained BERT model. Finally, we used 1, 2, and 0 as dataset outputs. Among the labels, "1" indicated positive sentiment, "0" indicated negative sentiment, and "2" indicated neutral sentiment (Table 2). Finally, the training and test datasets of the data were divided using Sklearn.

**Table 2.** Data sample.

| | Comments | Label |
|---|---|---|
| 1 | Very nice place, one side of the river is the remains of the Westerners, including churches, city gates, etc.; the other side of the river is the living area of the people, very sensational. | 1 |
| 2 | Walk by and feel free to look. No tickets! | 2 |
| 3 | Driving in Melaka is really uncomfortable, driving on the left is really not easy to grasp, and the traffic is too chaotic and there are many cars. | 0 |

To evaluate the application model, we determined the accuracy and performance evaluation indicators, namely Accuracy, Precision, Recall, and *F*1-Score. These four indicators are often used to measure the performance of applied models [48]. The four

indices were computed using the confusion matrix in Table 3 with the formula shown in Equations (1)–(4).

$$\text{Accuracy} = \frac{A + D}{A + B + C + D} \tag{1}$$

$$\text{Precision} = \frac{A}{A + C} \tag{2}$$

$$\text{Recall} = \frac{A}{A + B} \tag{3}$$

$$F1 = \frac{2 \times P \times R}{P + R} \times 100\% \tag{4}$$

**Table 3.** Classification confusion matrix.

|  | Positive (Actual) | Negative (Actual) |
|---|---|---|
| Positive (predicted) | *A* | *B* |
| Negative (predicted) | *C* | *D* |

For the processed corpus, the Keras framework was used to build the BERT pre-training model, set the corresponding parameters, build the model structure, and specify a sentence of 300 words for input into the model. The specific parameters were as follows: vector dimension = 768, sentence length = 300, learning rate = 0.01, and activation function = Softmax. The data were trained at 80% accuracy and tested at 20% accuracy. The data distribution for each label is presented in Table 4.

**Table 4.** Data partitioning.

| Label | Training Set | Test Set | Total |
|---|---|---|---|
| positive | 7170 | 1793 | 8963 |
| negative | 2612 | 653 | 3265 |
| neutral | 4262 | 1065 | 5327 |

Experiments were carried out on accuracy, precision, recall, and *F*1 values. The model evaluation indicators are listed in Table 5. For the sentiment analysis of the Chinese travel reviews about Melaka, the predictive accuracy of the BERT-based deep learning approach was 92.78%, outperforming the normal sentiment predictive performance varying between 0.70 and 0.79 [49].

**Table 5.** Performance of the BERT-based deep learning methods for tourism sentiment analysis.

|  | Accuracy | Precision | Recall | *F*1 |
|---|---|---|---|---|
| BERT-based approach | 92.78% | 92.26% | 90.23% | 91.65% |

Here, we also performed a comparative analysis between the BERT model and three other applied models, namely, CharCNN, LSTM, and BiLSTM. These three deep-learning models have been developed in recent years, and they all can perform sentiment analysis on text and have advantages and disadvantages. To determine which model was more suitable for tourism text evaluation, the comparative analysis was based on the obtained accuracy, precision, recall, and *F*1 scores of the applied models (Table 6).

**Table 6.** Comparative result analysis using different models.

| Evaluation Metrics | CharCNN | LSTM | BiLSTM | BERT |
|---|---|---|---|---|
| Accuracy | 79.89% | 82.46% | 86.51% | 92.78% |
| Precision | 78.60% | 80.67% | 84.35% | 92.26% |
| Recall | 75.63% | 81.75% | 82.73% | 90.23% |
| *F*1-Score | 76.12% | 83.76% | 85.36% | 91.65% |

Compared with the other three models, the BERT model obtained the highest accuracy, precision, recall, and *F*1 score. Therefore, the BERT model is more effective for classifying the sentiments of heritage tourism.

*3.5. Sentiment Feature Analysis*

To further analyze the sentiment preferences of Chinese tourists, we calculated the high-frequency words of each category separately based on the results of the BERT model analysis and presented them using a cluster analysis program called Gephi [50].

This part of the study is divided into two steps: (1) collecting all high-frequency words based on the BERT sentiment prediction results, and (2) performing co-occurrence analysis on high-frequency words. The Gephi program generated a sentiment image network. In the network, the nodes represented high-frequency words in tourists' comments, the size of the nodes indicated their importance, and the distance indicated closeness. Clusters (shown in the same color) referred to more frequent co-present words representing a closer relationship between the nodes. Using this, we could capture the connections between the most frequently mentioned words, and thus the content and structure of the many travel reviews related to tourists' emotions.

**4. Results**

According to the statistics of word frequency, the study captured 23,133 positive words and 7362 negative words. The numbers of positive, neutral, and negative comments were 8963, 2327, and 689, respectively, of which positive comments dominated (Table 4). With regard to overall feelings of enjoyment or disgust, Chinese tourists preferred the Stadthuys, the most famous tourist destination in Melaka, and representative scenic spots; maritime museums ranked the lowest (Table 7).

**Table 7.** Melaka scenic spots with emotional score summary.

| NO. | Scenic Spot | Positive Reviews (%) | Negative Reviews (%) |
|---|---|---|---|
| 1 | Stadthuys | 93.92% | 0.38% |
| 2 | Port Santiago | 89.97% | 1.36% |
| 3 | St. Paul's Church | 89.81% | 0.84% |
| 4 | Jonker Street | 87.95% | 1.99% |
| 5 | Melaka River | 85.79% | 1.52% |
| 6 | Dutch Square | 85.76% | 1.84% |
| 7 | Sultanate Palace | 83.22% | 1.86% |
| 8 | Strait Mosque | 82.30% | 0.98% |
| 9 | Maritime Museum | 67.68% | 3.42% |

*4.1. General Sentiment Analysis Results*

To further explore the sentiment characteristics of Chinese outbound tourists, this study conducted semantic network analysis using Gephi. The sentiment image of Chinese tourists included five clusters (Figure 3), of which "Melaka" was the largest. This network cluster included four main scenic spots, each forming a cluster that included representative landmarks. For example, Port Santiago and Castle, St. Paul's Church, the church/graveyard/city gate, Dutch Square, and Stadthuys. The Sultanate Palace and Imperial Palace formed the smallest cluster with only two elements. It can be seen that Chinese

tourists had a strong interest in specific tourist attractions, which was consistent with the [22] research results. Other clusters included the Strait Mosque with its beautiful sunset, Jonker Street with many Chinese elements, and the Melaka River with romantic feelings. These clusters represented scenic spots with the most significant impact on tourists [9]. The perception keywords of "worthy," "famous," "characteristic," and "beautiful" basically constituted the overall impression of tourists on Melaka tourism.

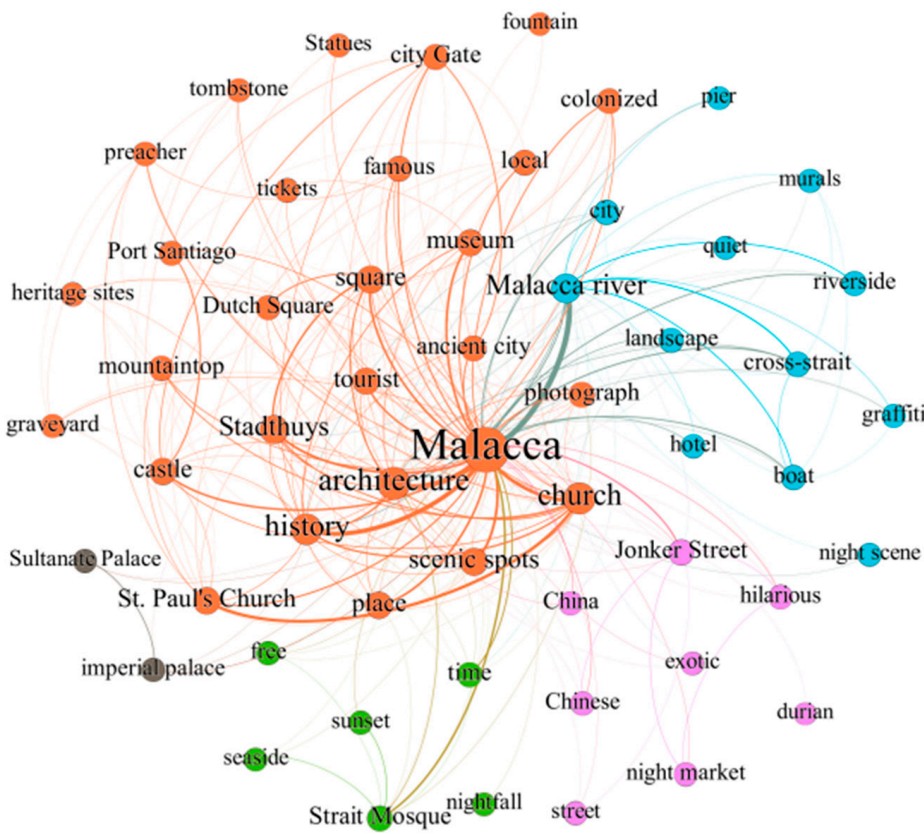

**Figure 3.** Sentiments image of general reviews.

In addition, as shown in Table 8, the keywords most frequently mentioned by Chinese tourists were architecture (31.25%), history (22.17%), heritage sites (11.70%), and landscapes (10.99%). This means that, among all tourist attractions in Melaka, Chinese tourists had the strongest feelings about architecture. By reviewing the review texts, we found that Chinese tourists were most interested in "architecture", not only because of its unique features and artistic value, but also because of its historical and religious cultural value. From the comments, it can be seen that most tourists had developed strategies before visiting and had some knowledge of the history and culture of these buildings, such as the architectural characteristics of the Stadthuys, the history and culture of Port Santiago, and the legends of St. Paul's Church. Second, there was the topic of "Heritage site", which highlighted the fact that Chinese tourists knew that Melaka is a heritage site before they went there and they were very interested in the heritage sites in Melaka, as shown by the sites with the most comments.

**Table 8.** Comments by identified and classified categories.

| Comments | Number | Percentage |
|---|---|---|
| Architecture | 2355 | 31.25% |
| Heritage site | 1671 | 22.17% |
| History | 882 | 11.70% |
| Landscape | 828 | 10.99% |
| Tourists | 735 | 9.75% |
| Ticket | 396 | 5.25% |
| Scenic spots | 354 | 5.22% |
| Sunset | 315 | 4.17% |
| Total | 7536 | 100% |

### 4.2. Positive and Negative Results

After identifying the general sentiments of Chinese tourists about Melaka, we extracted high-frequency words of positive and negative evaluations for each of the nine attractions, which were then analyzed separately to identify the emotional characteristics of Chinese tourists and to explore the causes of their positive and negative emotions.

The sentiment image for what was positively evaluated by Chinese tourists (Figure 4) was composed of five clusters. Its overall structure and elements were similar to those of the general sentiment network (Figure 3), showing a multicenter structure with heritage names as the core nodes of the subnetwork. In the first cluster, the central node of positive evaluation in the Gephi image was also "Melaka" and "architecture". The difference was that the "Jonker Street" and "Melaka River" clusters in the overall evaluation were replaced by "Port Santiago" and "Stadthuys" clusters. This showed that, although tourists had a high number of comments on Jonker Street and the Melaka River, their positive comments were relatively low. Chinese tourists had more positive feelings towards Port Santiago and Stadthuys.

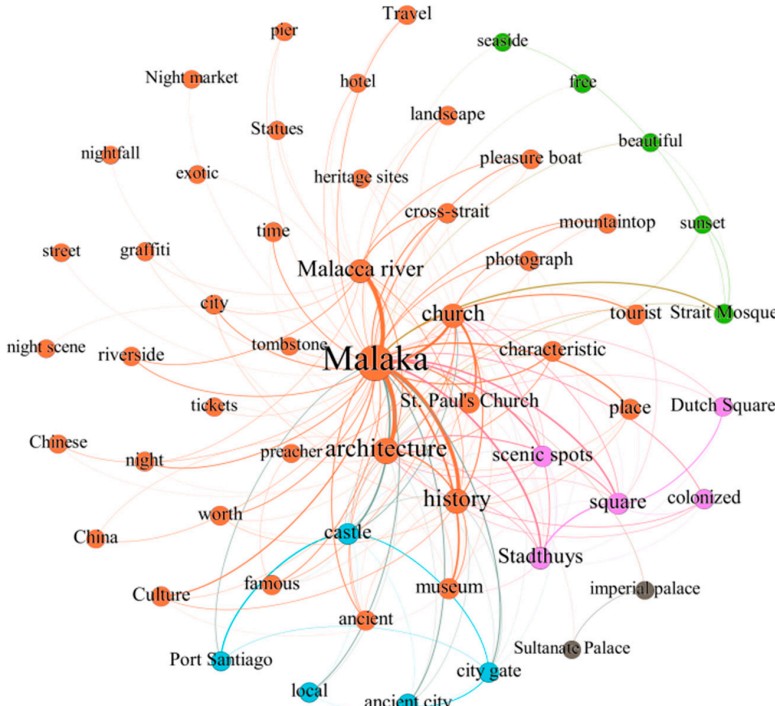

**Figure 4.** Sentiments image of the positive reviews.

Most Chinese tourists had a positive attitude towards the main tourist attractions in Melaka; positive comments included: (1) Words such as "history," "architecture," "ancient," and "culture," indicating that Chinese tourists appreciated the rich history of the local buildings in Melaka. The authenticity of pre-war buildings, sites, and monuments attracted

Chinese tourists, who felt a strong historical and cultural atmosphere at heritage sites. (2) The words "heritage" and "history" indicated that Chinese tourists paid close attention to heritage and were willing to care about, protect, and share cultural heritage. (3) The words "characteristic," "famous," and the like indicated that Chinese tourists were more interested in characteristic scenic spots of the heritage site, or scenic spots with high Internet recommendations, such as the "Dutch Red House," "St. Paul's Church", "Santiago Castle", etc. (4) The words "evening," "cross-strait," "scenery," and "sunset" indicated that the natural scenery of Melaka also attracted Chinese tourists. (5) The words "special features" and "snacks" indicated that Melaka's unique material heritage and "snacks" have attracted many Chinese tourists. When choosing food, tourists were keen on online celebrity food and the top dishes in the food recommendation software.

Negative comments are shown in Figure 5. Five semantic clusters were more likely to attract negative comments from tourists, but the node frequency was much lower, indicating that positive comments were more powerful than negative comments. The first cluster was centered on "architecture," including three scenic spots: Stadthuys, Dutch Square, and Sultanate Palace. Chinese tourists complained that Stadthuys and Dutch Square are crowded, and that the Sultanate Palace is "simple". The second cluster was centered on "church", with comments concentrated on the church in Port Santiago. Chinese tourists complained most about "dilapidation and "narrowness". The third cluster focused on scenic spots. Tourists complained about Melaka being too tiny, the dilapidated Port Santiago, and too many tourists; the fourth cluster was centered on the Melaka River. Tourists complained that the Melaka River emits a terrible smell. The fifth cluster was centered on the Strait Mosque. Tourists thought that it is not worth going into, but only suitable for taking pictures at sunset. It was not attractive to them. Overall, the negative comments mainly included: (1) "shabby," "narrow," and "simple". This showed that some tourists had negative comments about Melaka because the site is shabby and simple and the streets are too narrow. (2) Because Melaka is not large, the scenic spots are relatively concentrated. At small intersections, there are many landmark places such as Stadthuys, the Melaka River, and Jonker Street. Thus, tourists become too concentrated, which affects their experiences and leads to negative emotions. (3) Most of the attractions of Melaka that one can visit are architectural sites with single functions. If tourists are not interested in heritage tourism, they complain that it is boring.

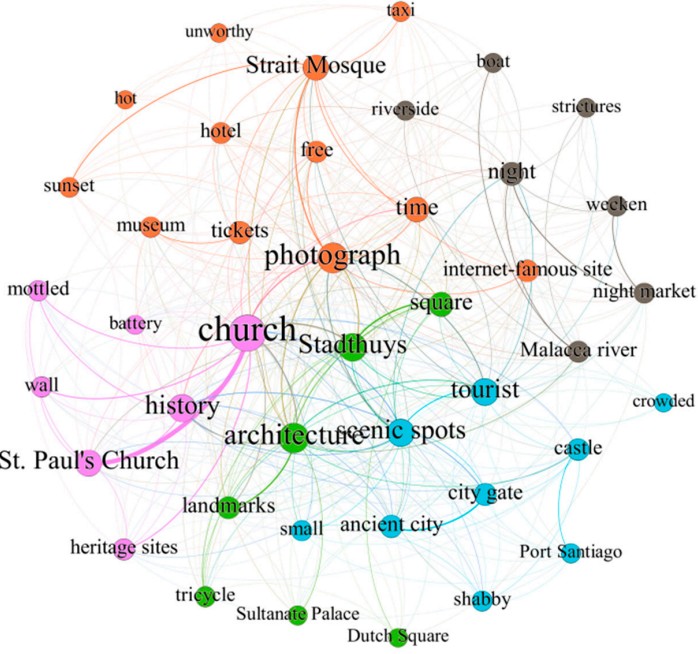

**Figure 5.** Sentiments image of the negative reviews.

## 5. Discussion

Based on the above conclusions, the heritage of the overall cultural atmosphere, material culture, and landscape characteristics are the main reasons for the positive evaluation of Chinese tourists. From the positive emotional words associated with "architecture," "history," "feature," "culture," and "heritage", it can be seen that Chinese tourists were very clear about the heritage identity of Melaka. Their tourism motivation involved visiting heritage sites in Melaka and the overall mood was positive. This was consistent with the research of [51], where Chinese tourists paid more attention to the attributes of cultural relics and historical sites and had a strong perception of the size and scale of the sites, the style and structure of buildings, etc. [9]. The positive emotional words "worthy" and "beautiful" were the high evaluation of the overall environment of the heritage site by Chinese tourists. Chinese tourists believed that the architectural sites and natural scenery in Melaka reflect the local characteristics.

Chinese tourists' negative perceptions of attractions, associated with words such as "architecture", "very small", "narrow" and "dilapidated", indicated that Chinese tourists felt pity for the small building area, small collections, dilapidation, and other problems in Melaka's scenic spots. In terms of their tourism experience, the Chinese tourists were very sensitive to crowding, and "too many people" appears in the high-frequency words many times. This was consistent with Hassan's (2014) study [52]. Chinese tourists were also highly sensitive to commercialization, consistent with Moy's (2014) study [53]. Through reviewing the comments related to the high-frequency words "uninteresting" and "unworthy," it was found that, although Chinese tourists had a strong cultural perception of the Melaka architectural sites, they had a weak perception of the deep historical background and cultural connotations. Their perceptions of heritage sites were intuitive and shallow [54]. Therefore, if tourists did not know the history of Melaka in advance, or were not interested in heritage tourism, they may have felt frustrated. The high-frequency word "culture" referred more to the historical culture and architectural cultures, and there was almost no comment on intangible culture. It can be inferred that Melaka tourism products were relatively few, and cultural experience interaction was less, which was also the main reason for the negative emotions. This is not similar to Moy's (2014) study [53], where Moy analyzed a questionnaire and concluded that the reason for Chinese tourists' negative feelings in Melaka were safety, discrimination against Chinese, artificial features of sites, hospitality and service to Chinese, transportation, visa ease, etc.

Generally, there were more positive evaluations than negative evaluations, which indicated that most Chinese tourists were satisfied with the heritage site. From the perspective of tourism demand, the primary purpose of Chinese tourists visiting Melaka was to experience rich and diverse foreign culture. This was consistent with the results of a study by Moy (2014) [53]. This cultural difference created a strong attraction for Chinese tourists. It was not only the motivation of Chinese tourists to travel but also a critical factor in meeting their tourism needs. Although most of the tourism experiences of Chinese tourists were only visual perceptions, they still fulfilled their tourism needs to seek novelty and differences, so they had a high degree of satisfaction. This was consistent with the findings of Liu (2018) [22].

Therefore, it was confirmed that sentiment analysis based on the BERT model can effectively analyze the emotional characteristics of tourists and the reasons for their positive and negative emotions. This type of analysis provides valuable information for improving the attractiveness of tourist destinations.

## 6. Conclusions and Implications

In this study, a sentiment analysis method was applied based on deep learning to investigate Chinese tourists' tourism perceptions of Melaka. To accomplish this research work, we mainly considered the BERT model. To understand the performance of the BERT in this context, we applied CharCNN, LSTM, and Bi-LSTM machine learning techniques also. Lastly, the obtained result of the BERT was compared with the other three applied

techniques in terms of performance evaluation techniques. It was found that BERT outperformed the other applied techniques in achieving an accuracy of 92.78%. Based on the BERT model, we conducted a sentiment analysis of tourism texts. It was found that Chinese tourists' emotional perceptions of Melaka were positive, and that they were very interested in heritage sites. The most important reason for the negative emotions of Chinese tourists was a lack of cultural experiences in Melaka, which made Chinese tourists feel "bored". In order to further enhance the tourism image of Melaka as a World Heritage Site, and to ensure that its history and culture are fully explored, thus achieving sustainable development of tourism for the heritage site, the following recommendations were made.

*6.1. Theoretical Implications*

The contributions of this study are as follows: First, it provides a valuable perspective for tourism perception research using BERT based on deep learning methods, which can reveal critical emotional characteristics and explain tourists' positive and negative views of a destination. Second, the BERT-based deep learning approach can serve as an interface between big data analysis and tourism management theory, which can help enrich our theoretical knowledge about tourists' perceptions and extend the current perception theory. Third, the analysis can help researchers understand the potential causes of tourists' positive and negative emotions, and it can suggest various action plans to improve the performance of tourism destination services.

*6.2. Management Implications*

After the adjustment of the Chinese government's epidemic policy on 8 January 2023, a large number of Chinese tourists started to travel abroad [7]. In order to better enhance the attractiveness of Melaka to Chinese tourists, this study proposes the following tourism policy recommendations:

1. During the peak tourism season, overcrowding, management confusion, and insufficient staff can occur easily. Heritage sites need to strengthen their management and supervision, such as posting peak hours on the website. This may alleviate overcrowding and visitor expectations. GIS-based mobile apps can help managers analyze visitors' spatio-temporal activities and predict potential congestion and congestion problems [55]. Destination sustainability could be improved by means of flow control to avoid overtourism [9].

2. To strengthen the unique tourism image of the destination, enhance competitiveness, attract tourists, and promote the sustainable development of the destination, first, as a world cultural heritage site, the government and related departments need to deeply explore the architectural culture and intangible cultural heritage, customs, and other cultural resources contained in multicultural integration, and accurately and deeply refine and publicize them. Enhancing Chinese tourists' experiences and perceptions of Melaka's unique architectural culture and historical stories will further enhance Chinese tourists' curiosity and yearning for foreign culture. Second, cultural identity and local attachment are essential factors for tourists' positive feelings. Through elements such as historical monuments, unique natural landscapes, and colorful folk activities, the cultural and spiritual connotations of heritage sites are fully integrated into tourism programs, enhancing the cultural atmosphere of heritage sites, which helps to stimulate tourists' cultural identity and local attachment, and strengthen the emotional connection between tourists and the heritage site.

3. Cultural performance is an intangible heritage covered by UNESCO's classifications. Multiculturalism can further enhance the background of Melaka to improve tourists' understanding of the multi-ethnic lineages that have been assimilated for more than 500 years, such as Baba Nyonya and Chetti [52]. Displaying this unique culture can further enhance heritage activities and experiences by providing local and international tourists with the same experiences. Through the ingenious design of various festivals and entertainment activities, scenic spots can be connected to drive the flow

of people through tourist attractions. Local residents can also be integrated into experiential projects that highlight the authenticity of local culture. For example, local residents can volunteer in scenic spots and memorials, or make handicrafts in intangible cultural experience halls. Through emotional exchanges between residents and tourists, tourists can come to deeply understand the local culture, it will improve their enthusiasm and participation, and it can make them ambassadors for the intangible cultural heritage.

4. To effectively protect the original cultural ecology of the region and the nation, and to explore the sustainable use of its heritage resources, both to meet the needs of the present generation and to protect the needs of future generations, heritage tourism in Melaka should be explored from a heritage protection perspective. The Melaka Tourism Administration should work closely with relevant conservation agencies to renovate and rebuild dilapidated sites and restore their former glory. To ensure that heritage sites can be restored and maintained, they should be used to generate a sustainable income.

## 7. Limitation and Future Research

Chinese tourists are among the most important sources of tourists in Malaysia. Research shows that they are less likely to participate in mass tourism activities but are more interested in tourism or travel that emphasizes an expanding experience. COVID-19 has had an immeasurable negative impact on social economy, personal consumption ability, and tourism development, especially for World Heritage Sites whose primary source of livelihood is tourism income. As the collection of online comments for this study ended in December 2019, before the COVID-19 pandemic, the current research lacks an understanding and insight into the impact of COVID-19 on Chinese tourists' experiences. Future research can study the behavioral changes of domestic and international tourists in Melaka, as well as the technological innovation and adaptation of heritage tourism, such as artificial intelligence and robots, to improve the experience of tourists under the new normal in the post-COVID-19 period.

Using the tourism comments of Chinese tourists, this study revealed the feelings and experiences of Chinese tourists. However, this study has some limitations. First, this study was based on user-generated online content, which may lead to bias and subjectivity in comment content, as users are more often from younger groups [42]. Future research could analyze comments from multiple sources. Second, the model used for text analysis was unable to identify sarcasm, jokes, or hyperbole; these sentiments were manually identified. Third, the study utilized comments posted between 2015 and 2019; future studies could compare these data with data from previous years. Additionally, management decisions cannot only rely on user-generated content. A hybrid method combining big data analysis and classic behavioral methods will be used to reveal the complete cognition of tourists, which is what will be carried out in the next step.

**Author Contributions:** Conceptualization, Z.C. and H.X.; methodology, Z.C.; software, Z.C.; formal analysis, Z.C.; writing—original draft preparation, Z.C. and H.X.; writing—review and editing, H.X.; visualization, Z.C.; supervision, B.S.-X.T. All authors have read and agreed to the published version of the manuscript.

**Funding:** This research was funded by the 2021 Henan Provincial Higher Education Institution Philosophy and Social Science Innovation Team "Research on the Integration of Ideological and Political Education and Innovation and Entrepreneurship Education in the New Era" (grant number 2021-CXTD-11).

**Institutional Review Board Statement:** Not applicable.

**Informed Consent Statement:** Not applicable.

**Data Availability Statement:** Not applicable.

**Conflicts of Interest:** The authors declare no conflict of interest.

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
