# Peer review of "Sentiment of Chinese Tourists towards Malaysia Cultural Heritage Based on Online Travel Reviews"

_sustainability, doi:10.3390/su15043478_

Round 1

Reviewer 1 Report

Dear Authors!

The manuscript is very interesting. Touches on a current topic.

The title reflects the content of the manuscript.

There is a balanced relationship between the purpose, content and conclusion of the manuscript.

The manuscript is well written and interesting.

The manuscript complies with the rules of academic ethics.

A small recommendation for a better understanding is to slightly improve the abstract.

The abstract must include sufficient information for readers to judge the nature and significance of the topic. The abstract should contain the main idea of the paper, the subject and the goal of the research, methods used, hypotheses, research results and a brief conclusion.

It is better to submit the text from the third person, not like you, in particular ("we analyzed", "we calculated"...). It is better to write: it is analyzed, it is calculated, etc.

I am grateful to the authors for the interesting material they have prepared.

Best regards

Reviewer 2 Report

The manuscript analyzed the role of the online travel reviews using a deep

learning approach to investigate Chinese tourists' perceptions of the heritage site in Malacca, to reveal the emotional characteristics of Chinese outbound tourists, and the factors influencing positive and negative emotions. The selected topic of the manuscript is of value both in theory and practice. The presentation is logical and scientific. However, some minor shortcomings should be addressed:

The article is too long. There are repetitions about BERT in sections where is not useful. Please check the flow of the text. 

abstract: Melaka? is the same Malacca or not?

BERT: you use BERT as well as acronym. Please explain or clear 

Literature review: lines 87-89 are not useful 

Study area: Please insert a map or a photograph of the location; p. 7 line 256 please explain RM 

Conclusion: p 16 lines 498-516 they are repetitions already told in the text 

Reviewer 3 Report

The authors deal with an essential topic of tourist sentiment in the context of cultural/heritage tourism. Indeed, it has been challenging to measure it using classical measurement methods. Using the deep learning method and tools is the novelty that the authors underline within the paper as its main strength. The authors put much effort into conducting their impressive research.

However, sentiment analysis based on tourist comments on social media or travel/tourist platforms, measured using web scraping techniques and analysed using advanced software, has already been employed by commercial travel market research agencies such as TCI Research, Data Appeal Company and others. Therefore, the manuscript will benefit if the authors explain the novelty of their research in the context of market development of the tourist sentiment area.

The authors would explain in a more detailed way in what areas deep learning methods bring novelty to tourist sentiment analysis. Now, only one paragraph is dedicated to this and does not bring. Similarly, the BERT model explanation should go further than suggesting that it "[c]an also extract relational features at several different levels, which reflect the sentence semantics more comprehensively." 

In the introduction, the authors might better justify the need for a deeper understanding of behavioural intentions at heritage sites instead of focusing on research gaps in Chinese tourism. Sustainability is an international journal so the audience would gain universal knowledge based on internationally recognised research gaps.

I can't entirely agree heritage tourism is a "niche area". On the contrary, I would say it is one of the significant tourist markets.

Considering that the use of deep learning methods for analysing tourist sentiment should be seen as a novelty and a significant methodological contribution of the paper. The authors stress, "The BERT model is the most suitable depth-learning model for tourist sentiment analysis". As such, in the conclusion section, I expect they indicate what results could be achieved using the BERT model compared to previously used sentiment analysis tools. Moreover, they should focus on the methodological implication of the study.

Putting the manuscript in the broader context of sustainability (as it is the scope of the Journal), the authors should justify how their approach leads to more sustainable use of heritage tourist resources.

On page 11, there is a technical comment. Please avoid leaving it before submitting the revision.

Round 2

Reviewer 3 Report

Thank you for the detailed and comprehensive reply to my remarks. We do not understand each other only in the first point, but it does not influence the general opinion about the revision.

Author Response

Thank you for your review!